# Heart rate phase is an indicator of chronotype-relevant circadian shifts associated with human disease: an *All of Us* Research Program analysis

Zachary M. Chan[1], Priya P. Patel[1], Gabriella M. Levitsky[1], Colleen A. McClung[2], Seyed Mehdi Nouraie[1,3] (ID) and Neil J. Kelly[1,4] (ID)

[1]*Department of Medicine, Center for Pulmonary Vascular Biology and Medicine, Pittsburgh Heart, Lung, and Blood Vascular Medicine Institute, University of Pittsburgh School of Medicine and University of Pittsburgh Medical Center, Pittsburgh, Pennsylvania, USA*
[2]*Translational Neuroscience, Department of Psychiatry, University of Pittsburgh, Pittsburgh, Pennsylvania, USA*
[3]*Division of Pulmonary, Allergy, Critical Care Medicine, and Sleep Medicine, Department of Medicine, University of Pittsburgh School of Medicine and University of Pittsburgh Medical Center, Pittsburgh, Pennsylvania, USA*
[4]*Department of Medicine, VA Pittsburgh Healthcare System, University of Pittsburgh, Pittsburgh, Pennsylvania, USA*

Handling Editors: Karyn Hamilton & Bettina Mittendorfer

The peer review history is available in the Supporting Information section of this article (https://doi.org/10.1113/JP290337#support-information-section).

**Abstract figure legend** Heart rate phase (HRP) was quantified from wearable-derived heart rate measurements. Earlier HRP was associated with pregnancy-associated conditions, while later HRP was associated with addiction and mood, metabolic, and sleep-related disorders in a phenome-wide analysis. Mendelian randomization analysis suggested a possible causal link between the morningness genomic variant rs1144566 and protection from type 2 diabetes mellitus through HRP.

The Journal of Physiology

**Abstract** Chronotype describes an individual's day or night preference and is thought to exert health effects through circadian timing, yet continuous circadian measures related to chronotype have never been examined in large human cohorts. We sought to quantify wearable-derived heart rate phase (HRP) as an indicator of chronotype-relevant circadian shifts and determine its associations with human disease. Heart rate was aggregated over 5 min intervals in *All of Us* participants with wearable data and fit to a sine curve with 24 h period to determine HRP. Associations between HRP and circadian genomic variants were determined by linear regression. A phenome-wide association study (PheWAS) was performed by multiple logistic regression. One-sample Mendelian randomization (MR) was performed by two-stage residual inclusion. Average HRP was $9.48 \pm 1.57$ h ($n = 15,960$). PheWAS identified phenome-wide associations of later HRP with addiction, mood, sleep and metabolic disorders, while certain conditions of pregnancy were associated with earlier HRP. In focused analyses of type 2 diabetes mellitus (T2DM), later HRP was associated with increased T2DM risk [odds ratio (OR) 1.09 [1.06,1.13], $P = 1.58 \times 10^{-7}$]. The morningness genomic variant rs1144566(T) was associated with earlier HRP ($P = 1.38 \times 10^{-7}$) and decreased risk of T2DM (OR 0.69 [0.55,0.88], $P = 2.67 \times 10^{-3}$). MR analysis suggested a causal effect of rs1144566 on T2DM risk through HRP ($P = 2.70 \times 10^{-3}$). We introduce the use of a quantitative longitudinal circadian metric, HRP. We combine HRP with phenomics, genomics and electronic health records data to provide evidence for a relationship between circadian shifts and disease. HRP may hold value in the management and surveillance of human health.

(Received 25 October 2025; accepted after revision 27 February 2026; first published online 18 March 2026)

**Corresponding author** N. J. Kelly: Pittsburgh Heart, Lung, and Blood Vascular Medicine Institute, University of Pittsburgh, School of Medicine and UPMC, 1723 Biomedical Science Tower, 200 Lothrop Street, Pittsburgh, PA 15261, USA. Email: njk88@pitt.edu

**Key points**

- Chronotype is a genetically influenced innate preference for the timing of daily activity and is believed to affect health outcomes through shifts in circadian rhythms.
- Measurement of chronotype-relevant circadian changes may be possible through wearable technology.
- We show that the timing of daily heart rate rhythms, which we call heart rate phase (HRP), is associated with chronotype-relevant genetic and demographic factors.
- We report associations of HRP with addiction, mood, sleep and metabolism disorders as well as certain pregnancy conditions.
- We show that HRP is associated with type 2 diabetes mellitus (T2DM) risk and that a morning genetic chronotype variant may causally influence T2DM risk through HRP.

## Introduction

Chronotype refers to an individual's innate preference for the timing of daily activities and has been associated with human health and disease (Adan et al., 2012; Jones et al., 2019). The health effects of chronotype are believed to be mediated through circadian rhythms, which refer to approximately 24 h physiological oscillatory rhythms

**Zachary Chan** is a junior at Fox Chapel Area High School in Pittsburgh, Pennsylvania, USA. Inspired by his early morning workouts as captain of the varsity rowing team, Zachary sought opportunities to research the health impacts of morning shifts in circadian rhythms. Zachary aspires to attend college in the future where he plans to study science and engineering.

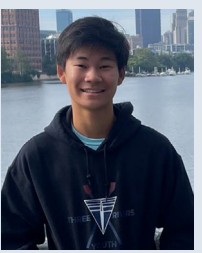

and are known to shift in accordance with chronotype class (Benloucif et al., 2005; Gibertini et al., 1999). However, direct measurement of circadian rhythms can be a burdensome undertaking involving serial blood draws (Benloucif et al., 2005), which is neither feasible nor practical for large-scale long-term studies of chronotype. As a result, there is a lack of population research on the health impacts of shifted circadian rhythms. Hence, there exists a significant need for scalable measures of shifted circadian rhythm in health research.

Chronotype is generally assessed by validated surveys such as the widely used Morningness–Eveningness Questionnaire (MEQ), which classifies respondents on a discrete scale from morning to neutral to evening types (Horne & Ostberg, 1976), with neutral types comprising roughly 60% of the population (Adan et al., 2012). Chronotype and circadian rhythm are closely related, as exemplified by the morning phase shift of the circadian melatonin rhythm in morning versus evening chronotypes (Gibertini et al., 1999). In contrast to chronotype, circadian timing exists on a continuous spectrum, and the relationship between chronotype and circadian timing is known to vary within chronotypes and with demographic variables such as age (Duffy et al., 1999). Given their mutability through entrainment and other means (Duffy & Wright, 2005), shifts in circadian rhythms may be more clinically actionable metrics than innate chronotypes.

Chronotype is also subject to strong genetic influences, as illustrated in multiple genome-wide association studies (Hammerschlag et al., 2017; Hu et al., 2016; Jones et al., 2016, 2019). In meta-analyses (Jones et al., 2019), a locus on chromosome 1 spanning the *RGS16* and *RNASEL* genes has shown the strongest associations with self-reported chronotype. Additionally, chronotype associations with variants in core clock genes including *CLOCK*, *PER* and *BMAL* have been reported (Kalmbach et al., 2017). While circadian rhythms have not been investigated in the majority of these variants, it is notable that a well-studied chronotype-associated variant in *PER3* has been reported to have a negligible effect on circadian timing as estimated by plasma melatonin (Hasan et al., 2012). Taken together, these data suggest an inconsistent relationship between chronotype and circadian rhythm as reflected by serial blood draws. Consequently, the lack of a more sensitive and real-time measure of shifted circadian rhythm may mask clinically meaningful associations.

Given the known rhythmicity of daily heart rates and the widespread adoption of wearable technology, we postulated that the phase of heart rate rhythms can fulfil the systematic need for a continuous, objective and scalable metric of circadian timing. The *All of Us* Research Program (*AoU*), which hosts commercial wearable data from more than 50,000 participants, offers a novel platform to interrogate the interplay between circadian rhythms, genomics and health outcomes. Using the *AoU* resource, we demonstrate the rhythmicity of longitudinal heart rate monitoring, the association of chronotype genomic variants with heart rate phase (HRP) and the unbiased phenome-wide association of HRP with chronotype-associated health conditions. As a model chronotype-associated disease, we demonstrate the relationship between HRP and type 2 diabetes mellitus (T2DM). Furthermore, we utilize genomic chronotype to offer evidence that chronotype may act causally through HRP to influence disease risk.

## Methods

### Ethical approval

The NIH *All of Us* (*AoU*) Institutional Review Board (IRB) serves as the single IRB for the use of *AoU* data. The *AoU* IRB reviews the protocol, informed consent and participant-facing materials for the *AoU* Research Program and follows regulations and guidance set out by the Office for Human Research Protections. The study conformed to the standards set by the *Declaration of Helsinki*, except for registration in a database

### Study period

Fitbit data were collected between 5 December 2014 and 1 October 2023. Electronic health records (EHR) data were available through 1 October 2023.

### Phase calculation

Mean participant heart rate (HR) was aggregated over 5 min intervals for each weekday (Sunday to Sunday) for all available participant dates, resulting in a series of 2016 time points (288 time points per day for 7 days). The resulting series was fit to a sine curve with 24 h (1440 min) period of the form $HR = A \sin(\frac{\pi}{720}(t - \varphi)) + b$ where $A$ represents amplitude, $t$ time in minutes, $\phi$ heart rate phase (abbreviated HRP), and $b$ vertical shift using the Python package *lmfit* v1.3.4. Participants with HR data on $\leq 30$ days or sine curve goodness-of-fit ($R^2$) $\leq 0.5$ were excluded ($n = 21{,}978$ participants). The optimal $R^2$ threshold was determined by the maximum Youden index, which we computed as the participant retention rate plus the $R^2$ threshold minus 1; the participant retention rate was defined as the number of participants in the Fitbit cohort at a given $R^2$ threshold divided by the number of participants in the Fitbit cohort at an $R^2$ threshold of 0.0. To allow for continuous analysis of HR phase data, participants with phases $\geq 3$ standard deviations beyond the mean were excluded ($n = 286$ participants).

## Sleep duration

Total sleep duration was calculated as the sum of light, deep and rapid eye movement (REM) sleep duration as described previously (Chan et al., 2025). Average sleep duration was calculated as the average total sleep duration across all participant-nights.

## Single nucleotide variants

Single nucleotide variants (SNVs) previously associated with circadian rhythm (Gene Ontology GO:0007623, $n = 122$ SNVs in *AoU* genomics data) were obtained from the publicly available Genome-Wide Association Study (GWAS) Catalog (Cerezo et al., 2025). Genotypes, sex ploidy and genomic ancestry were obtained from the *AoU* workbench (Bick et al., 2024). Allele frequencies (AFs) were calculated for each SNV allele based on frequencies in the genotyped *AoU* population ($n = 414,830$ participants). For each SNV, the two most common alleles were studied; the reference (non-effect) allele was defined as the most frequent allele, and the alternate (effect) allele as the second most frequent. Twelve SNVs with alternate allele AF < 1% were excluded, with 110 SNVs remaining in the analysis.

## Genotype–phase association

HRP was modelled as a function of genotype, age, sex ploidy (modelled as a binary variable reflecting the presence or absence of a Y chromosome), and the first five principal components of predicted genomic ancestry using ordinary least squares regression with robust standard errors in *statsmodels* v0.14.2 (Seabold & Perktold, 2010). Genotypes were modelled additively as the number of copies of the alternate allele. Participants whose SNV genotype included alleles other than the two most common alleles were excluded from the analysis of that SNV. Age was computed as the mean age on dates for which HR data were available. Adjusted *P*-values were computed by the Bonferroni method. An adjusted *P*-value <0.05 was considered statistically significant. $R^2$ and $D'$ calculations were performed with the LDlink webtool using GRCh38 and all populations (Machiela & Chanock, 2015).

## Phenome-wide association study

*AoU* conditions are annotated using the Observed Medical Outcomes Partnership (OMOP) common data model (Voss et al., 2015), which is organized as a hierarchical tree. Participants were considered to have a given condition if they had an associated occurrence of either the condition or its 1st or 2nd degree descendants. The phenome-wide association study (PheWAS) was limited to two degrees of separation in order to identify the most significant levels of the hierarchical tree. Conditions present in more than 100 participants were considered. Logistic regression was performed to model the presence or absence of each identified condition as a function of HRP, weekly steps, the lesser of age at death or end of the study period, sex (presence of a Y chromosome) and the first five principal components of genomic ancestry prediction. Analyses were performed using the *logit* function of the Python package *statsmodels* v0.14.2 with robust standard errors. Adjusted *P*-values were computed by the Bonferroni method. An adjusted *P*-value <0.05 was considered statistically significant.

## T2DM diagnoses

Participants were considered to have T2DM if they had any associated occurrences of T2DM (OMOP concept ID 201826) or any of its descendants. Odds ratios (ORs) for T2DM were computed from logistic regression as a function of function of HRP or rs1144566 genotype, weekly steps, the lesser of age at death or at the end of the study period, sex (presence of a Y chromosome) and the first five principal components of genomic ancestry prediction. Logistic regression was performed as described above. Linearity of log-odds was confirmed by data visualization.

## Haemoglobin A1c

Haemoglobin A1c (HbA1c, OMOP concept ID 3004410) levels were obtained from the *AoU* workbench. HbA1c measurements within the range of 1–30% were included. To exclude the effect of anti-hyperglycaemic medications on lower HbA1c, each participant's maximum HbA1c before the end of the Fitbit monitoring period was considered. Participants with no available HbA1c measurement in the time period were excluded. An HbA1c >5.6% was considered elevated (Echouffo-Tcheugui & Selvin, 2021). ORs for elevated A1c were computed from logistic regression as a function of HRP or rs1144566 genotype, weekly steps, age at the end of the Fitbit monitoring period, sex (presence of a Y chromosome) and the first five principal components of genomic ancestry prediction. Logistic regression was performed as described above. Linearity of log-odds was confirmed by data visualization.

## Cox proportional hazards

Age at diagnosis was evaluated by Cox proportional hazards using *CoxPHFitter* in the Python package

*lifelines* v0.30.0 as a function of HRP or rs1144566 genotype, average weekly steps and the first five principal components of genomic ancestry prediction. Analyses were stratified by sex (presence of a Y chromosome) to satisfy the proportional hazards assumption. The proportional hazards assumption was tested using the *check_assumptions* feature of *lifelines*. Age at diagnosis was considered as the earliest condition occurrence of T2DM (OMOP concept ID 201826) or any of its descendants. Participants were right censored at the lesser of age at death or age at the end of the study period.

### Mendelian randomization analysis

Mendelian randomization (MR) was performed using the two-stage residual inclusion (2SRI) method (Gorfine et al., 2024). In the first stage, using ordinary least squares, the endogenous variable (HRP) was regressed on the instrumental variable (additive rs1144566 genotype) and the exogenous variables average weekly steps, age, sex and the first five principal components of the genomic ancestry prediction. In the second stage, using logit, the outcome variable (diagnosis of T2DM or A1c > 5.6%) was regressed on the residual from stage 1, HRP and the above exogenous variables. Regression analyses were performed in the *statsmodels* Python module. Stage 2 confidence intervals were determined by bootstrapping with 100,000 resamples.

### Statistics

Locally estimated scatterplot smoothing (LOESS) regression curves and 95% confidence intervals were generated using the Python module *scikit-misc* v0.5.1 with a smoothing factor of 0.1. Polar plots and circular statistics were generated using the Python module *pycircstat2* v0.1.12. Regression, kernel distribution estimate (KDE) and violin plots were generated using the Python module *seaborn* v0.12.2. Survival curves were generated in the Python *lifelines* module. LOESS regression, Forest plots volcano plots and Manhattan plots were generated in Graphpad PRISM v10.4.2 (Dotmatics, Boston, MA, USA). Comparisons of continuous variables between two groups were made using a two-tailed independent sample Welch's *t* test for normally distributed data or Mann–Whitney U test for non-normally distributed data in *scipy* v1.11.2 (Virtanen et al., 2020). Determination of normality was made by visual inspection of data distributions and Q–Q plots generated in *scipy*. Effect sizes and confidence intervals between two groups were estimated by Cohen's *d* and calculated using the Python module *pingouin* v0.5.5.

## Results

### Participant selection and background characteristics

We identified *AoU* participants who voluntarily shared Fitbit recordings of heart rate ($n = 53,686$ participants). A flow diagram of participant selection is shown in Fig. 1*A*. Only participants with greater than 30 days of HR data ($n = 48,629$ participants) were considered. Average heart rate over 5 min intervals was mapped to the corresponding time of the week, beginning Sunday at midnight, and fit to a sine curve with 24 h (1440 min) period to determine HRP. Participants with sine fit $R^2 > 0.5$ and HRP within 3 SD of the population mean HRP were included for further analyses ($n = 31,422$ participants); the threshold of $R^2 > 0.5$ was selected using Youden's index to balance dual interests in participant retention and goodness of fit (Fig. 1*B*). Subsets of this group had accompanying whole genome sequencing data (WGS; $n = 22,653$ participants), and a further subgroup granted access to electronic health record data (WGS + EHR; $n = 15,690$ participants).

Participant demographics and HR phase characteristics are shown in Table 1 for the entire cohort as well as the WGS and WGS + EHR subsets. The mean age was $53 \pm 16$ years. Participants recorded an average of $54,400 \pm 24,800$ steps per week. Among participants with available sequencing data, Y chromosomes were present in approximately one-third of the population, and individuals of predicted European ancestry accounted for greater than 75% of the study population.

### HRP is associated with chronotype genomic variants

Among the Fitbit population meeting inclusion criteria ($n = 31,422$ participants), average HRP was $9.48 \pm 1.57$ h (Fig. 2*A–B*). To test the association between HRP and chronotype, we quantified its relationship with genomic variants previously linked to circadian rhythm traits in the WGS cohort ($n = 22,653$ participants). In linear regression models adjusted for age, sex ploidy and predicted genomic ancestry, we identified seven variants associated with HRP (Fig. 2*C–D*, Table S1). Of these variants, the four most significant – rs1144566, rs684383, rs516134 and rs12736689 – were in high linkage disequilibrium in a 20 kb region near the *RGS16* and *RNASEL* genes (Table S2). This locus has consistently shown the most robust associations to morning chronotype in prior GWAS and meta-analyses (Jones et al., 2019). Taken together, these data support an association between HRP and chronotype variants.

### HRP is inversely correlated with weekly steps counts and age

For the remainder of our analyses, we utilized the WGS + EHR cohort ($n = 15,690$) containing wearable,

genomic and health data. Given the associations between HRP and chronotype variants, we sought first to determine whether HRP was associated with physical activity and age (Adan et al., 2012; Fischer et al., 2017; Polanska et al., 2024), which have also been linked to chronotype. We compared average weekly step counts to HRP and observed a significant correlation between later HRP and reduced steps (Pearson $r = -0.3107$, $P < 0.0001$, Fig. 3A), paralleling reported associations between evening chronotype and reduced step counts. We also observed a correlation between earlier HRP and increased age (Pearson $r = -0.2409$, $P < 0.0001$, Fig. 3B), consistent with prior literature that chronotype advances during adulthood (Adan et al., 2012; Fischer et al., 2017). The relationship between chronotype and sex is complex and age-dependent (Fischer et al., 2017), and we observed no significant difference in HRP between sexes as defined by the presence or absence of a Y chromosome [Cohen's $d = 0.02$, 95% confidence interval (CI) [$-0.02,0.05$], $P = 0.335$ Fig. 3C]. Taken together, our reproduction of known associations provides additional support for the use of HRP as a quantitative circadian indicator relevant to chronotype.

## PheWAS identifies significant HRP associations with health conditions

We next sought to determine the landscape of health conditions associated with HRP in an unbiased manner, hypothesizing that they would overlap with chronotype-associated conditions. In logistic regression models adjusted for weekly steps, age, sex and genomic ancestry, we identified significant associations between HRP and the odds of 22 of 3272 prevalent or historical conditions. We classified these conditions into addiction & mood disorders, sleep disorders, metabolic disorders and pregnancy-related conditions (Fig. 3D, Table S3). Our findings of increased risk of addiction & mood (Logan et al., 2014), sleep (Partonen, 2015) and metabolic (Feng et al., 2025; Fu et al., 2023; Hashemipour et al., 2020; Peng et al., 2022; Reutrakul et al., 2013; Romanenko et al., 2024) disorders with later HRP are broadly consistent

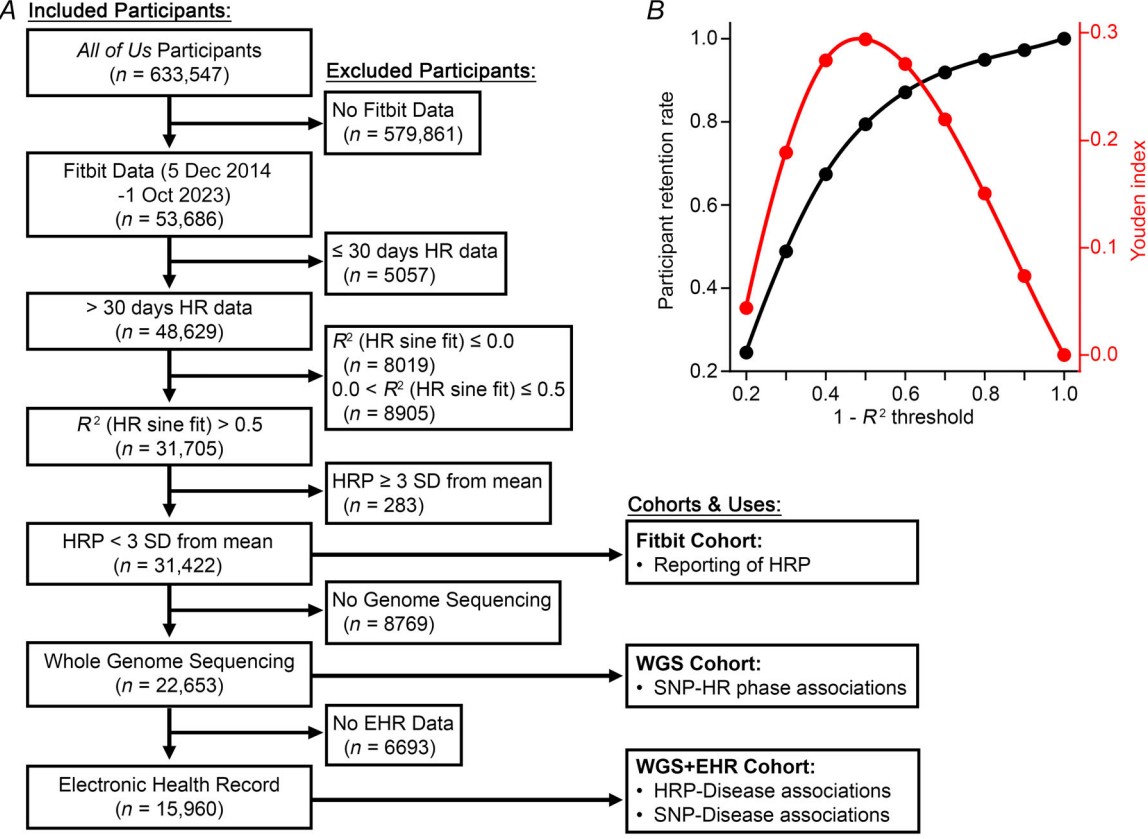

**Figure 1. Participant selection strategy**
*A*, flow diagram of participant selection. *B*, participant retention rates at $R^2$ threshold increments of 0.1 (black, left axis) were quantified as the Fitbit cohort size at each $R^2$ threshold divided by the Fitbit cohort size at an $R^2$ threshold of 0.0. Youden index (red, right axis) was calculated as participant retention rate plus $R^2$ threshold minus 1 at each $R^2$ threshold increment.

**Table 1. Participant characteristics**

|  | HR | WGS | WGS + EHR |
|---|---|---|---|
| *n* | 31,422 | 22,653 | 15,960 |
| Mean age (years) | 53.19 ± 16.35 | 53.89 ± 16.39 | 54.49 ± 16.30 |
| Y chromosome | – | 7411 (32.72%) | 5145 (32.24%) |
| Genomic ancestry |  |  |  |
| amr | – | 2348 (10.37%) | 1547 (9.69%) |
| eur | – | 17,354 (76.61%) | 12,310 (77.13%) |
| afr | – | 1617 (7.14%) | 1218 (7.63%) |
| sas | – | 434 (1.92%) | 264 (1.65%) |
| eas | – | 816 (3.60%) | 564 (3.53%) |
| mid | – | 84 (0.37%) | 57 (0.36%) |
| Heart rate |  |  |  |
| HRP (h) | 9.48 ± 1.57 | 9.44 ± 1.56 | 9.43 ± 1.55 |
| $R^2$ | 0.74 [0.65,0.82] | 0.74 [0.65,0.81] | 0.74 [0.65,0.82] |
| No. of days | 344 [96,943] | 296 [91,989] | 296 [91,992] |
| Steps (10,000/week) | 5.43 ± 2.49 | 5.47 ± 2.49 | 5.44 ± 2.48 |

Columns show participants used for determination of HRP (HR column), SNV associations (WGS) and EHR associations (WGS + EHR). Data are presented as mean ± SD, median [interquartile range] or *n* (%).
Abbreviations: amr: admixed American, eur: European, afr: African, sas: South Asian, eas: East Asian, mid: Middle Eastern, $R^2$: HR sine curve goodness-of-fit.

with prior literature on evening chronotype. Meanwhile, certain conditions of pregnancy were more common with earlier HRP despite sex adjustment, consistent with prior hypotheses that morning chronotype is associated with greater fertility (Toffol et al., 2013). In summary, these phenome-wide findings suggest the utility of HRP as a clinically meaningful indicator associated with chronotype.

### HRP delay is associated with T2DM risk and onset

Given our PheWAS findings, we next sought to demonstrate the utility of HRP as a physiological marker in the context of a chronotype-relevant disease. Because T2DM is strongly associated with chronotype (Hashemipour et al., 2020; Reutrakul et al., 2013), highly prevalent (Chew et al., 2023), presents during adulthood (Le et al., 2021) and has a common associated laboratory metric in haemoglobin A1c (HbA1c), we selected T2DM for additional analysis. For all analyses, we performed adjustments for weekly steps, age, sex and the first five principal components of genomic ancestry prediction. We first confirmed an association between T2DM or any of its descendant conditions (*n* = 2161 of 15,960 participants) with later HRP (OR = 1.09 [1.06,1.13], $P = 1.58 \times 10^{-7}$, Fig. 4*A*, Table S4). Next, we utilized HbA1c measurements as a second method of assessing diabetes risk. To control for the probable dynamic qualities of HRP and the effects of anti-diabetic medications, we considered only the maximum recorded HbA1c prior to the end of the Fitbit monitoring period. Of 6439

participants with qualifying HbA1c measurements, 3392 had an abnormally elevated HbA1c defined as greater than 5.6% (Echouffo-Tcheugui & Selvin, 2021). In a multivariate logistic regression model, we found a significant association between later HRP and risk of elevated HbA1c (OR = 1.07 [1.03,1.11], $P = 3.48 \times 10^{-4}$, Fig. 4*B*, Table S5). Finally, we reasoned that participants with greater diabetes risk would be diagnosed at a younger age, and we hypothesized that participants with later HRP would be diagnosed with T2DM earlier. Indeed, in sex-stratified Cox proportional hazards models, we found that later HRP was significantly associated with younger age at T2DM diagnosis [hazard ratio (HR) = 1.17 [1.14,1.21], $P = 1.32 \times 10^{-28}$, Fig. 4*C–E*, Table S6]. This relationship was maintained in Cox proportional hazards models limited to the subset of participants diagnosed with T2DM after the initiation of heart rate monitoring (HR = 1.20 [1.12,1.28], $P = 1.86 \times 10^{-7}$, Fig. 4*F–H*, Table S7), suggesting that HRP–T2DM associations are present prior to the time of diagnosis. Collectively, these findings from activity-adjusted analyses suggest that circadian timing may independently influence the risk of T2DM.

### Morningness genotype is associated with protection against diabetes risk and onset

To validate the link between chronotype and HRP, we next sought to determine whether morning chronotype exerts its effects on T2DM risk through circadian timing as indicated by HRP. In the absence of survey-based data on chronotype, we selected rs1144566 as a genetic marker

of chronotype. In our analyses, rs1144566 had the most significant (lowest *P*-value and greatest magnitude) HRP association within the widely validated *RGS16-RNASEL* locus (see Table S2) and overall. In addition, it has been identified as a chronotype-associated variant in multiple prior GWAS and meta-analyses (Jones et al., 2019). As expected, among all participants with the two most common variants (C or T) at rs1144566 ($n = 15,923$ participants with only C or T alleles of 15,960 participants in the WGS + EHR cohort), the presence of a T allele was associated with significantly earlier HRP (Cohen's $d = 0.16$, 95% CI [0.09,0.23], $P = 3.39 \times 10^{-6}$, Fig. 5*A*), while rs1144566 genotype was not significantly associated with average weekly step counts (Cohen's $d = 0.05$, 95% CI [−0.02,0.12], $P = 0.314$, Fig. 5*B*). We confirmed that the odds of T2DM at any time (OR = 0.69 [0.55,0.88], $P = 2.67 \times 10^{-3}$, Fig. 5*C*, Table S8) and odds of elevated HbA1c prior to the end of the Fitbit monitoring period (OR = 0.75 [0.58,0.96], $P = 0.0230$, Fig. 5*D*,

Table S9) were lower in logistic regression models of risk as a function of the number of rs1144566 T alleles. In addition, in sex-stratified Cox proportional hazard models, rs1144566(T) was associated with a significant delay in the age at T2DM diagnosis (HR = 0.72 [0.58,0.89], $P = 3.20 \times 10^{-3}$, Fig. 5*E–G*, Table S10).

## MR suggests a causal link between rs1144566 genotype and T2DM risk

Finally, based on our findings that rs1144566(T) is associated with HRP and T2DM risk, we hypothesized that this genomic variant acts as an instrumental variable to causally increase T2DM risk through HRP (Fig. 6*A*). To test this hypothesis, we performed one-sample MR analysis using the 2SRI method. We found significant associations with HRP in models of T2DM (*F*-statistic = 399.88, $R^2 = 0.19$, $P = 2.70 \times 10^{-3}$, Fig. 6*B*,

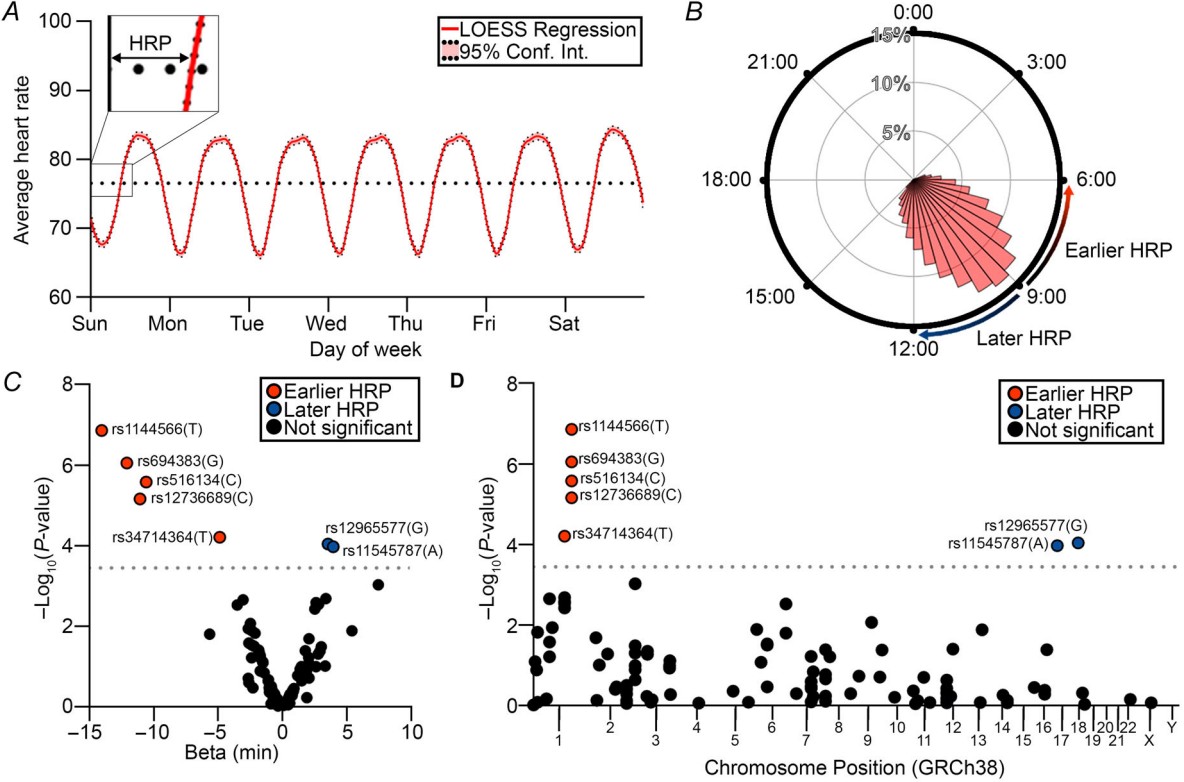

**Figure 2. Development and validation of HRP as a chronotype-related measure of circadian shift**
Average heart rate for each participant ($n = 31,422$ participants) was aggregated and averaged over 5 min intervals mapped to time of week. *A*, the population heart rate average was smoothed by LOESS regression for visualization. Heart rate *versus* time was fit to a sine curve with 24 h period, and the resultant heart rate phase (HRP, *A* inset) was plotted (*B*) on a polar histogram annotated with concepts of 'earlier' and 'later'. Among biallelic SNVs previously associated with circadian rhythm, multivariate linear regression was performed to quantify the allelic effect on HRP as a function of genotype in the population subset with WGS data ($n = 22,653$ participants). *C* and *D*, SNV–HRP associations were plotted as a volcano plot showing effect size in minutes (*C*) and Manhattan plot showing chromosomal location (*D*), where dashed horizontal line shows Bonferroni-adjusted $P < 0.05$. Linear regressions were adjusted for age, presence of Y chromosome and the first five principal components of genomic ancestry prediction.

Table S11) and elevated HbA1c ($F$-statistic $= 151.32$, $R^2 = 0.19$, $P = 0.0216$, Fig. 6$C$, Table S12) risk. These data offer evidence that rs1144566 genotype may play a causal role in T2DM risk through HRP.

## Discussion

In this study, we define HRP as a chronotype-related metric of circadian shifts and demonstrate its association with human disease. Utilizing a large and diverse cohort of participants with paired wearable, genomic and health data, we reported concordant findings between HRP and previous chronotype-based reports through multiple methods including relevant genomic variants and demographics. We performed a PheWAS to identify

significant associations between HRP and addiction, mood, sleep, metabolic and pregnancy-associated conditions. Finally, we utilized knowledge of genomic chronotype associations to show that a morning chronotype variant is associated with and may offer direct protection against T2DM through HRP.

Because of the continuous and scalable metric that HRP offers, a PheWAS of chronotype and circadian timing was made possible in this study. In doing so, we identified 22 phenome-wide significant associations with HRP after step-count adjustment. These findings are consistent with prior targeted studies of disease–chronotype associations, showing that phase delays in HRP are associated with an increased risk of metabolic (Feng et al., 2025), addiction & mood (Logan et al., 2014) and sleep disorders

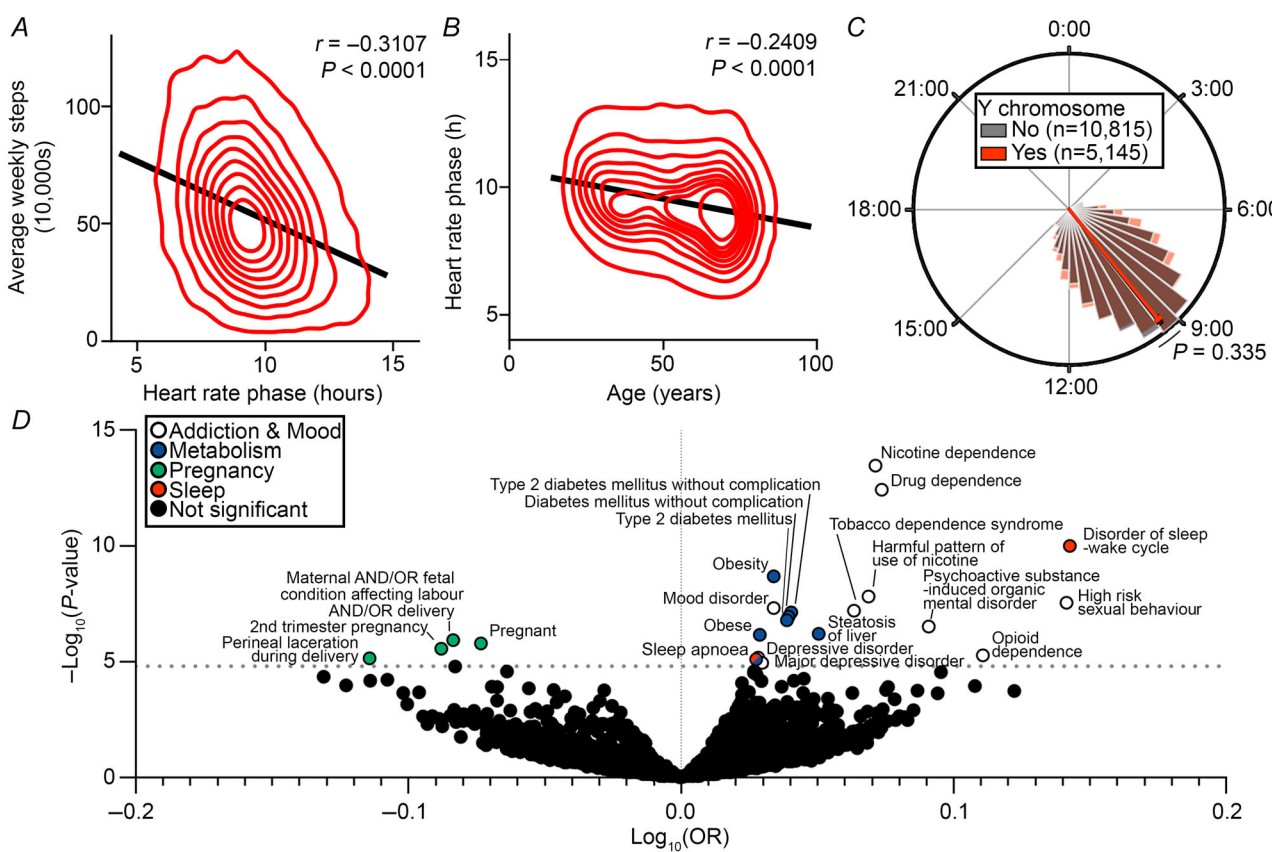

**Figure 3. HRP is associated with addiction, mood, sleep, metabolism and pregnancy-related conditions in PheWAS**

*A* and *B*, HRP was compared to weekly steps (*A*) and age (*B*) on distribution plots with linear regression (*n* = 15,960 participants). *C*, HRP distributions between participants with and without a Y chromosome were plotted in polar histograms (*n* = 5145 participants with Y chromosome and *n* = 10,815 participants without Y chromosome). *D*, odds ratios (OR) of any documented prevalent or historical condition as a function of HRP were calculated in a phenome-wide association study by multivariate logistic regression adjusted for weekly steps, age, presence of a Y chromosome and the first five principal components of genomic ancestry prediction (*n* = 15,960 participants); horizontal dashed black line represents Bonferroni-adjusted *P* < 0.05. Distribution plots show kernel density estimate (red) and univariate linear regression (black) where *r* represents the Pearson correlation coefficient and its associated *P*-value. Polar plots show a histogram of HRP values with buckets spanning 30 min intervals. Comparisons between two groups were made by independent two-sided Welch's *t* test. *P* < 0.05 was considered statistically significant.

(Partonen, 2015). Interestingly, our finding that phase advances in HRP are associated with an increased odds of pregnancy-associated conditions in a sex-adjusted model offers additional support to speculated links between morningness and fertility (Toffol et al., 2013).

The finding of activity-adjusted HRP-disease associations brings additional attention to the potential benefits of timed activity beyond the volume of activity. Notably, morning exercise has been reported to confer disproportionate health benefits over evening

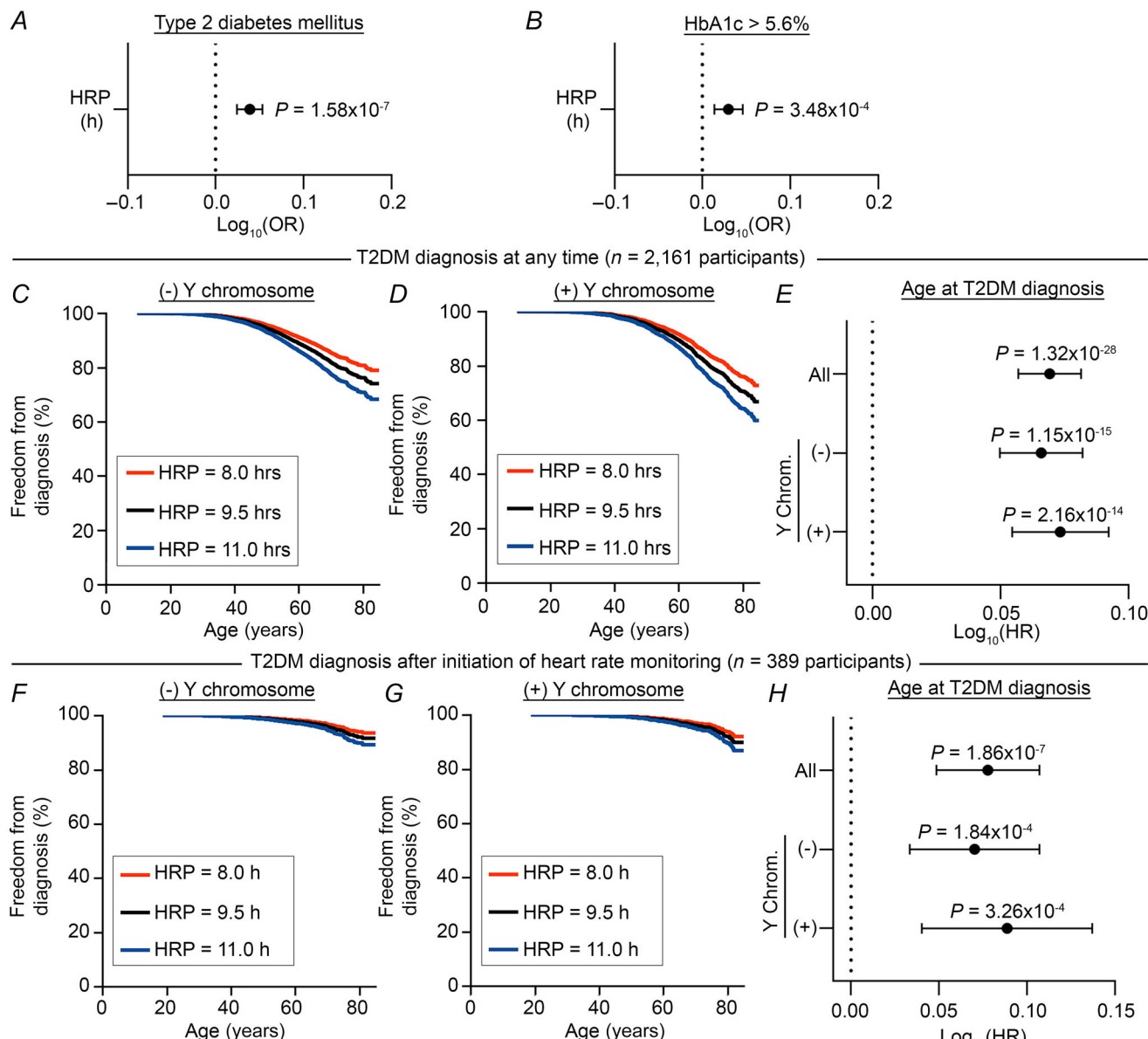

**Figure 4. Evening HRP is associated with increased risk of T2DM**
*A* and *B*, ORs of T2DM (*A*; *n* = 15,960 participants) and elevated HbA1c (*B*; *n* = 6439 of 15,960 participants with qualifying HbA1c measurements) as a function of HRP were calculated by multivariate logistic regression. *C, D* and *E*, hazard ratio (HR) of age at T2DM (*n* = 2161 participants with T2DM of 15,960 participants) as a function of HRP was calculated by Cox proportional hazards stratified by the absence (*C*) or presence (*D*) of a Y chromosome. *F, G,* and *H*, subgroup analysis was performed on participants diagnosed with T2DM after the initiation of heart rate monitoring (*n* = 389 participants), and HR of age at T2DM as a function of HRP was calculated by Cox proportional hazards stratified by the absence (*F*) or presence (*G*) of a Y chromosome. Proportional hazards were adjusted for weekly steps and the first five principal components of genomic ancestry prediction; multivariate regression analyses were additionally adjusted for age and presence of a Y chromosome. Forest plot error bars show 95% confidence interval. *P* < 0.05 was considered statistically significant.

exercise (Rynders & Broussard, 2024), and it is possible that morning exercise acts coordinately on metabolic–circadian axes to provide dual advantages. In fact, the rhythmic phases of heart rate and activity are closely linked (Natarajan et al., 2025); if the benefits of morning exercise are achieved partially through advancement of circadian phase, one could speculate whether natural or pharmacological interventions to achieve phase advancement are sufficient to confer a degree of protection against HRP-associated diseases.

In our disease-focused analyses, we chose to investigate HRP relationships with T2DM primarily due to its high prevalence and onset during adulthood, which coincides with the adult demographic of the *All of Us* cohort (Le et al., 2021). In step-adjusted analyses, we demonstrate consistent associations between phase-delay in HRP and increased risk of T2DM and elevated HbA1c as well as earlier age at T2DM diagnosis. In our analysis of the morningness variant rs1144566(T), we find that the morning allele is associated with decreased risk of those

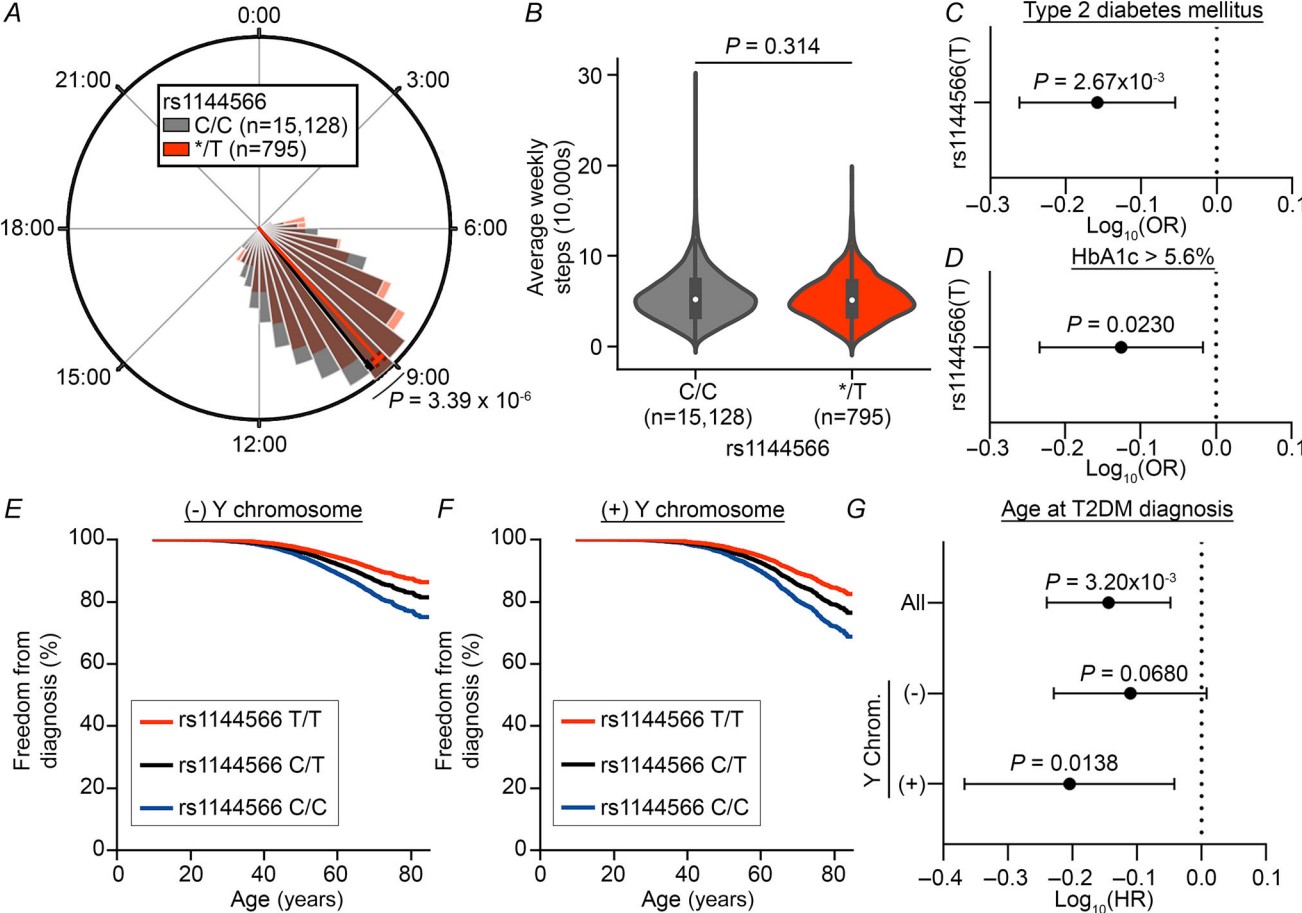

**Figure 5. The rs1144566(T) genotype is associated with protection from T2DM**
Participants harbouring only T or C alleles at rs1144566 were evaluated (*n* = 15,923 of 15,960 participants). *A* and *B*, HRP (*A*) and weekly steps (*B*) were compared between participants with (*/T, *n* = 795 participants) or without (C/C, *n* = 15,128 participants) an rs1144566 T allele. *C* and *D*, odds ratios (OR) of T2DM (*C*; *n* = 15,923 participants) or elevated HbA1c (*D*; *n* = 6420 of 15,923 participants with qualifying HbA1c measurements) as a function of rs1144566 genotype were calculated by multivariate logistic regression. *E–G*, hazard ratio (HR) of age at T2DM diagnosis as a function of rs1144566 genotype (*n* = 2155 participants with T2DM of 15,923 participants) was calculated by Cox proportional hazards stratified by the absence (*E*) or presence (*F*) of a Y chromosome (*G*). Polar plots show histogram of HRP values with buckets spanning 30 min intervals; buckets with fewer than 20 participants were pooled and averaged with adjacent buckets until the total number was greater than 20. Violin plots show kernel density estimate with overlayed box plot where the white dot represents the median, box represents interquartile range and bars represent 1.5× the interquartile range. Comparisons between two groups were made by an independent two-sided Welch's *t* test (*A*) or Mann–Whitney U test (*B*). Proportional hazards were adjusted for weekly steps and the first five principal components of genomic ancestry prediction; regression analyses were additionally adjusted for age and presence of a Y chromosome. Forest plot error bars show 95% confidence interval. *P* < 0.05 was considered statistically significant.

three T2DM phenotypes. We also conducted MR analysis to determine whether a causative relationship exists between rs1144566(T) and T2DM risk. Due to the underlying statistical assumptions, caution is warranted in the inference of causality from our monogenic one-sample MR (Burgess et al., 2023). It is possible that rs1144566 genotype also influences T2DM risk through other means; notably, rs1144566 encodes a missense variant in *RGS16*, which has been shown to regulate insulin secretion and pancreatic beta-cell proliferation *in vitro* (Vivot et al., 2016). Yet, RGS16 is also thought to play a significant role in the maintenance of central circadian rhythms (Doi et al., 2011). Taken together, these data suggest that RGS16 may protect against T2DM via multiple mechanisms. In contrast to our findings, prior MR analyses did not find associations between morningness genotypes and risk of T2DM (Jones et al., 2019). However, these polygenic analyses did not probe continuous and scalable measures of circadian timing such as HRP (Jones et al., 2016, 2019). As such, we expect that polygenic MR analyses will be further refined to control for pleiotropic effects as larger study cohorts emerge where circadian timing is available (Burgess et al., 2023).

A major limitation of our study is the absence of self-reported chronotype data in the *AoU* dataset. While we offer several lines of evidence to demonstrate similarities between HRP and chronotype at the levels of genomics, demographics and disease, future studies are required to directly determine this relationship. However, one could argue that HRP has greater health relevance than chronotype, as it is a continuous and scalable measure of circadian rhythm, which is thought to serve as the intermediary between chronotype and health (Benloucif et al., 2005; Gibertini et al., 1999). Furthermore, the mutable nature of circadian rhythm, as opposed to intrinsic chronotype, makes it an attractive target for health interventions.

Our association study of circadian variants identified significant associations between HRP and seven circadian rhythm-associated SNVs. Importantly, the directionality of HRP shift was congruent with odds of morning or evening chronotype reported previously (Hammerschlag et al., 2017; Hu et al., 2016; Jones et al., 2016, 2019). The absence of significant associations with most reported circadian variants has several possible explanations. First, our association study was limited to 22,653 participants, while previous survey-based GWAS have included hundreds of thousands of participants (Jones et al., 2019). Second, some chronotype-associated variants do not affect circadian rhythms or do so only under specific circumstances (e.g. in younger age groups; Adan et al., 2012); similarly, some chronotype-associated variants may manifest in rhythms other than heart rate. Additionally, the diverse ancestral makeup of our cohort may mask population-specific chronotype associations, as prior GWAS consisted of individuals with European ancestry (Hu et al., 2016; Jones et al., 2016, 2019). Our study replicated associations with the *RGS16-RNASEL* genomic locus, which has been most strongly and consistently associated with chronotype (Jones et al., 2019).

We acknowledge additional limitations to this study. First, our findings remain associative and require validation in additional cohorts. Next, our regression analyses may be limited by the effects of unmeasured confounders. While we add the important contribution of step-count adjustments in evaluating the associations between circadian rhythm and disease, we did not consider the type and intensity of activity. Additionally, chronotype is not believed to play a major role in average sleep duration, although evening types may accumulate a greater sleep debt during the week with compensatory increases in weekend sleep duration (Adan et al., 2012; Jones et al., 2019). For this reason, we did

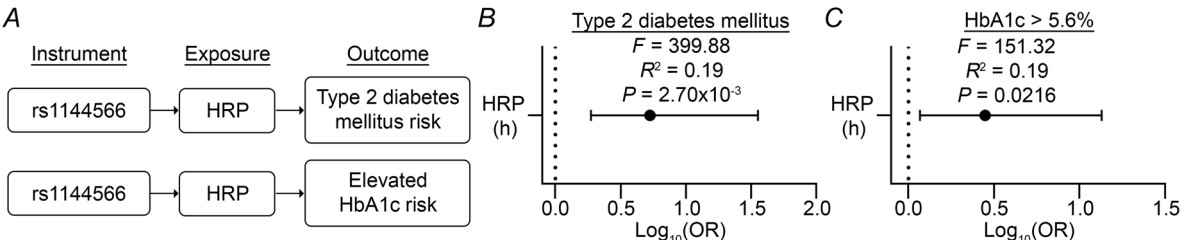

**Figure 6. Mendelian randomization analyses suggest a causal association between rs1144566 and T2DM through HRP**
*A*, directed acyclic graphs (DAGs) show the hypothesized role of rs1144566 as an instrument in the effect of HRP exposure on type 2 diabetes mellitus or elevated HbA1c outcomes. *B* and *C*, instrumental variable analysis was performed by two-stage residual inclusion (2SRI) for the exposure of HRP in type 2 diabetes mellitus (*B*; $n = 15,923$ participants) and elevated HbA1c (*C*; $n = 6420$ of 15,923 participants with qualifying HbA1c measurements) outcomes. Regression analyses were adjusted for weekly steps, age, presence of a Y chromosome and the first five principal components of genomic ancestry prediction. Forest plot error bars show the 95% confidence interval. *F*-statistic and $R^2$ are shown from Stage 1 of 2SRI. $P < 0.05$ was considered statistically significant.

not adjust for sleep duration in our disease association models. However, inclusion of average sleep duration as an additional covariate did not change the statistical conclusions of the study (Tables S13–S21). The interfaces of HRP with sleep architecture and sleep irregularity warrant further examination, particularly given the known links between sleep variability and human health (Zheng et al., 2024). Future studies should also investigate how HRP relates to indices of sleep timing, as irregular sleep–wake timing has previously been linked to impaired metabolic health (Taylor et al., 2016). Finally, we were unable to adjust for nutritional factors, which are broadly associated with metabolic disease and have been reported to vary with circadian rhythms (Reutrakul et al., 2013; van der Merwe et al., 2022).

In conclusion, we present here the results of a novel and phenome-wide approach to the study of circadian biology in disease using wearable technology. In tandem with genomic and EHR data, we identify HRP as a chronotype-associated circadian indicator that is associated with the risk of addiction, mood, sleep, metabolic and pregnancy-related conditions. Consequently, these results may contribute to novel approaches for the surveillance and maintenance of human health.

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

## Additional information

### Data availability statement

AoU data and code are available to authorized users of the All of Us Research Program's Controlled Tier Dataset 8 on the Researcher Workbench at https://www.researchallofus.org/.

### Competing interests

N.J.K. holds a research grant from United Therapeutics.

### Author contributions

Z.M.C., P.P.P. and N.J.K. conceived the project. Z.M.C., G,M.L. and N.J.K. wrote the manuscript. Z.M.C., P.P.P., G.M.L., S.M.N. and N.J.K. analysed the data and prepared the figures. C.A.M. and N.J.K. supervised the project. All authors reviewed, edited and approved the manuscript.

### Funding

This work was supported by the WoodNext Foundation (N.J.K. and C.A.M.), NIH grants T32 HL129964 (N.J.K.) and K08 ES037420 (N.J.K.), the United Therapeutics Jenesis Innovative Research Award (N.J.K.), and the Pulmonary Hypertension Association (N.J.K.).

### Acknowledgements

We gratefully acknowledge *All of Us* participants for their contributions, without whom this research would not have been possible. We also thank the National Institutes of Health's *All of Us* Research Program for making available the participant data examined in this study. This work is the result of NIH funding, in whole or in part, and is subject to the NIH Public Access Policy. Through acceptance of this federal funding, the NIH has been given a right to make the work publicly available in PubMed Central.

### Keywords

circadian rhythm, diabetes mellitus, metabolism

## Supporting information

Additional supporting information can be found online in the Supporting Information section at the end of the HTML view of the article. Supporting information files available:

**Peer Review History**
**Supporting Information**

