## [Peer Review History · The Journal of Physiology]

Heart rate phase is an indicator of chronotype-relevant circadian shifts associated with human disease: An *All of Us* research program analysis

Zachary M Chan, Priya P Patel, Gabriella M Levitsky, Colleen A McClung, Seyed Mehdi Nouraei, and Neil J Kelly
DOI: 10.1113/JP290337

Corresponding author(s): Neil Kelly (njk88@pitt.edu)

The following individual(s) involved in review of this submission have agreed to reveal their identity: Sean Williams (Referee #3)

Review Timeline:	Submission Date:	25-Oct-2025
	Editorial Decision:	16-Jan-2026
	Revision Received:	12-Feb-2026
	Accepted:	27-Feb-2026

Senior Editor: Karyn Hamilton

Reviewing Editor: Bettina Mittendorfer

Transaction Report:

Dear Dr Kelly,

Re: JP-RP-2025-290337 "**Heart rate phase is an indicator of chronotype-relevant circadian shifts associated with human disease: An *All of Us* research program analysis**" by Zachary M Chan, Priya P Patel, Colleen A McClung, Seyed Mehdi Nouraei, and Neil J Kelly

Thank you for submitting your manuscript to The Journal of Physiology. It has been assessed by a Reviewing Editor and by 2 expert referees and we are pleased to tell you that it is acceptable for publication following satisfactory revision.

REVISION CHECKLIST:

Please upload two versions of your manuscript text: one with all relevant changes highlighted and one clean version with no changes tracked. The manuscript file should include all tables and figure legends, but each figure/graph should be uploaded as separate, high-resolution files. The journal is now integrated with Wiley's Image Checking service. For further details, see: <https://www.wiley.com/en-us/network/publishing/research-publishing/trending-stories/upholding-image-integrity-wileys->

image-screening-service

We look forward to receiving your revised submission.

Yours sincerely,

Karyn Hamilton
Senior Editor
The Journal of Physiology

REQUIRED ITEMS

- You must start the Methods section with a paragraph headed Ethical Approval. If experiments were conducted on humans, confirmation that informed consent was obtained, preferably in writing, that the studies conformed to the standards set by the latest revision of the Declaration of Helsinki and that the procedures were approved by a properly constituted ethics committee, which should be named, must be included in the article file. If the research study was registered (clause 35 of the Declaration of Helsinki), the registration database should be indicated, otherwise the lack of registration should be noted as an exception (e.g. The study conformed to the standards set by the Declaration of Helsinki, except for registration in a database). For further information see: <https://physoc.onlinelibrary.wiley.com/hub/human-experiments>.

- Please upload separate high-quality figure files via the submission form.

- Your paper contains Supporting Information of a type that we no longer publish, including supplementary tables and figures. Any information essential to an understanding of the paper must be included as part of the main manuscript and figures. The only Supporting Information that we publish are video and audio, 3D structures, program codes and large data files. Your revised paper will be returned to you if it does not adhere to our Supporting Information Guidelines.

- Papers must comply with the Statistics Policy: https://jp.msubmit.net/cgi-bin/main.plex?form_type=display_requirements#statistics.

In summary:

- If $n \leq 30$, all data points must be plotted in the figure in a way that reveals their range and distribution. A bar graph with data points overlaid, a box and whisker plot or a violin plot (preferably with data points included) are acceptable formats.

- If $n > 30$, then the entire raw dataset must be made available either as supporting information, or hosted on a not-for-profit repository, e.g. FigShare, with access details provided in the manuscript.

- 'n' clearly defined (e.g. x cells from y slices in z animals) in the Methods. Authors should be mindful of pseudoreplication.

- All relevant 'n' values must be clearly stated in the main text, figures and tables.

- The most appropriate summary statistic (e.g. mean or median and standard deviation) must be used. Standard Error of the Mean (SEM) alone is not permitted.

- Exact p values must be stated. Authors must not use 'greater than' or 'less than'. Exact p values must be stated to three significant figures even when 'no statistical significance' is claimed.

- Please include an Abstract Figure file and an Abstract Figure legend. An appropriate figure legend, which should not exceed 150 words in length, should be included in the main manuscript file. The Abstract Figure is a piece of artwork designed to give readers an immediate understanding of the research and should summarise the main conclusions. If possible, the image should be easily 'readable' from left to right or top to bottom. It should show the physiological relevance of the manuscript so readers can assess the importance and content of its findings. Abstract Figures should not merely recapitulate other figures in the manuscript. Please try to keep the diagram as simple as possible and without superfluous information that may distract from the main conclusion(s). Abstract Figures must be provided by authors no later than the revised manuscript stage and should be uploaded as a separate file during online submission labelled as File Type 'Abstract Figure'. Please also ensure that you include the figure legend in the main article file. All Abstract Figures should be created using BioRender. Authors should use The Journal's premium BioRender account to export high-resolution images. Details on how to use and access the premium account are included as part of this email.

EDITOR COMMENTS

Reviewing Editor:

Comments for Authors to ensure the paper complies with the Statistics Policy :
Please follow the guidance provided by the Statistics Editor.

REFEREE COMMENTS

Referee #2:

This is a novel analysis of the 'all of us' data examining whether HRP can be used as an indicator of chronotype circadian shifts and the possible link to human disease.

This is an original use of wearable data to address a complex issue.

Sleep variability (time to bed and fall asleep) more than sleep duration has been linked with poor sleep. Why was this variable not used instead of sleep duration? Did many people have insomnia?

What was the age range of your subjects? How many were postmenopausal? Could menopause have caused a shift in HRP?

L176 How frequently was HbA1C measured?

L224 What was the range for number of steps?

L246 What is the correlation? P value?

L302 The decrease in weekly steps? Decreased from what? This is a bit confusing.

Since you were interested in the relationship to diseases why was CV disease not analyzed?

L387 Not using sleep variability should be included in the limitations, when sleep variability is linked with many diseases....maybe more so than sleep duration.

Referee #3:

Comments for Author (Required):

The manuscript employs independent sample t-tests for between-group comparisons (e.g., HRP by sex, genotype) but does not report assessment of normality assumptions (Shapiro-Wilk test, Q-Q plots) or homogeneity of variances (Levene's test). While the large sample sizes ($N > 20,000$) may justify invoking the Central Limit Theorem for robustness to non-normality, this should be explicitly stated. Additionally, given the substantially unequal group sizes in some comparisons (e.g., rs1144566 C/C $n=21,498$ vs */T $n=1,108$), Welch's t-test would be more appropriate than Student's t-test, or a non-

parametric alternative such as Mann-Whitney U should be considered. Crucially, with such large sample sizes, statistical significance alone is uninformative as trivial effects will be detected as significant. The authors should report effect sizes (e.g., Cohen's d) and confidence intervals alongside p-values to allow readers to assess the meaningfulness of reported differences.

The sine curve fitting methodology excludes approximately 35% of participants using an R^2 threshold of 0.5, which is not clearly justified. Sensitivity analyses using alternative thresholds and comparison of demographic characteristics between included and excluded participants would strengthen confidence that selection bias has not been introduced. Regarding the Mendelian randomisation analysis, I have limited experience with this form of analysis, but my understanding is that there should be mention of instrument strength metrics such as the F-statistic (ideally >10) or conditional F if adjusted for covariates. The authors acknowledge limitations of the single-SNV approach but may wish to expand on instrument validity.

Please ensure adherence to the Journal's policy on 'n' values:

All 'n' values must be clearly stated main text, figures and their legends or tables with 'n' clearly defined (e.g. x cells from y slices in z animals) in each location. Authors should be mindful of how 'n' is defined to avoid pseudoreplication.

END OF COMMENTS

Response to Referee Comments

EDITOR COMMENTS

Reviewing Editor:

Comments for Authors to ensure the paper complies with the Statistics Policy: Please follow the guidance provided by the Statistics Editor.

We thank the reviewing editor for the helpful feedback, which we believe has contributed to a substantially improved manuscript. Please see our specific point-by-point responses to the referees below. Line numbers reference the “clean” version of the manuscript.

REFEREE COMMENTS

Referee #2:

This is a novel analysis of the 'all of us' data examining whether HRP can be used as an indicator of chronotype circadian shifts and the possible link to human disease.

This is an original use of wearable data to address a complex issue.

We thank the reviewer for these encouraging comments and important recommendations. Please see our specific point-by-point responses below. Line numbers reference the “clean” version of the manuscript.

Comment 1: Sleep variability (time to bed and fall asleep) more than sleep duration has been linked with poor sleep. Why was this variable not used instead of sleep duration? Did many people have insomnia?

Response 1: We greatly appreciate this reviewer’s comment on the importance of sleep timing and its variability in overall sleep hygiene. In order to avoid collinearity in our regression analyses, we did not combine metrics of sleep timing and HRP, as both relate to circadian timing. However, we have revised our discussion to emphasize the importance of including sleep timing in future studies.

Regarding insomnia, there were 2,579 of 15,960 participants in the WGS+EHR cohort with a diagnosis of insomnia (see **Table S3**). There was a nominally significant association between insomnia and later HRP (OR 1.03 [1.00, 1.07], nominal *P*-value = 0.0368) which was not statistically significant after Bonferroni adjustment.

Comment 2: What was the age range of your subjects? How many were postmenopausal? Could menopause have caused a shift in HRP?

Response 2: Thank you for raising this important point, which we believe refers to the finding of an association between earlier HRP and pregnancy-related conditions in the phenome-wide association study (PheWAS). While accurate determination of menopausal status could be challenging in the *All of Us* cohort, the mean age of the population (53.19 ± 16.35 years, see **Table 1**) roughly approximates the average age of menopause in the United States. We have updated the results section of the text to include standard deviation in addition to mean age (line

236). **Figure 1B** illustrates a linear relationship between HRP and age in this predominantly (67%) female cohort, and age was included as a covariate in the PheWAS. Additionally, the PheWAS was constructed to determine the odds ratio of ever having a given condition (such as pregnancy) as opposed to having a condition only during the Fitbit monitoring period. We have revised the text to indicate that the PheWAS studied “prevalent or historical conditions” (line 272). Taken together, while it is likely that HRP differs between pre- and post-menopausal populations, it is unlikely to have influenced the results of the PheWAS.

Comment 3: L176 How frequently was HbA1C measured?

Response 3: Given the observational and retrospective nature of the study, there was no uniformity in HbA1c measurement frequency across the population. The mean number of available HbA1c measurements prior to the end of Fitbit monitoring in the EHR + WGS cohort was 2.3 measurements per participant. Among individuals who had at least one HbA1c measurement during that period, the mean number of measurements per participant was 5.7.

Comment 4: L224 What was the range for number of steps?

Response 4: The mean and standard deviation for weekly step counts are reported in **Table 1**. In the WGS+EHR cohort, the mean and standard deviation of weekly step count was 54,400 ± 24,800. We have updated the text of the results to include mean and standard deviation (line 237).

Comment 5: L246 What is the correlation? P value?

Response 5: We apologize for the lack of clarity in reporting of correlation statistics. We have now included the Pearson correlation coefficients and *P*-values in the text (lines 258, 260) as well as the accompanying figures.

Comment 6: L302 The decrease in weekly steps? Decreased from what? This is a bit confusing.

Response 6: We thank the reviewer for pointing out the confusing phrasing of this statement. In our re-analysis of the step comparisons between genotypes using a non-parametric test as recommended by **Referee #3**, we found no statistically significant difference in step counts. Because the statement no longer applies, we have removed this sentence from the manuscript.

Comment 7: Since you were interested in the relationship to diseases why was CV disease not analyzed?

Response 7: We apologize for any confusion regarding the PheWAS results. For this analysis, associations between HRP and all possible conditions (including CV) were analyzed (see **Table S3**). However, only conditions reaching phenome-wide significance were annotated in **Figure 3D**. For example, “bradycardia” was associated with HRP (odds ratio 0.9179 [0.8698, 0.9685]) with a nominal *P*-value of 0.0018 which was not statistically significant after Bonferroni adjustment.

Comment 8: L387 Not using sleep variability should be included in the limitations, when sleep variability is linked with many diseases....maybe more so than sleep duration.

Response 8: As suggested here and in **Comment 1**, we have updated the discussion to include additional emphasis on sleep timing and sleep variability. The limitations section of the discussion now reads (line 415):

“The interfaces of HRP with sleep architecture and sleep irregularity warrant further examination, particularly given the known links between sleep variability and human health (Zheng *et al.*, 2024). Future studies should also investigate how HRP relates to indices of sleep timing, as irregular sleep-wake timing has previously been linked to impaired metabolic health (Taylor *et al.*, 2016).”

Referee #3:

We thank the statistical reviewer for these important recommendations to improve the manuscript. As mentioned below in **Comment 12**, we have now limited all analyses in **Figures 3-6** to the EHR+WGS cohort (n=15,960 participants). This change applies to **Figure 3A-C** and **Figure 5A-B** only, such that these panels now include the same cohorts as the accompanying panels. Line numbers reference the “clean” version of the manuscript.

Comment 9: The manuscript employs independent sample t-tests for between-group comparisons (e.g., HRP by sex, genotype) but does not report assessment of normality assumptions (Shapiro-Wilk test, Q-Q plots) or homogeneity of variances (Levene's test). While the large sample sizes ($N > 20,000$) may justify invoking the Central Limit Theorem for robustness to non-normality, this should be explicitly stated. Additionally, given the substantially unequal group sizes in some comparisons (e.g., rs1144566 C/C n=21,498 vs */T n=1,108), Welch's t-test would be more appropriate than Student's t-test, or a non-parametric alternative such as Mann-Whitney U should be considered. Crucially, with such large sample sizes, statistical significance alone is uninformative as trivial effects will be detected as significant. The authors should report effect sizes (e.g., Cohen's *d*) and confidence intervals alongside p-values to allow readers to assess the meaningfulness of reported differences.

Response 9: We appreciate this very important comment regarding statistical analyses and reporting pertaining to two-sample comparisons (**Figure 3C** comparing HRP between sexes, **Figure 5A** comparing HRP between rs1144566 genotypes, and **Figure 5B** comparing step counts between rs1144566 genotypes). Because of the limitations of statistical tests of normality in large sample sizes, we did not perform Shapiro-Wilk or other similar tests. However, based upon your advice and following further inspection of the data distributions and Q-Q plots, we now use Welch's t-test for HRP comparisons between sexes (**Figure 3C**) and rs1144566 genotypes (**Figure 5A**). We use Mann-Whitney U test for comparison of step counts between genotypes (**Figure 5B**), which resulted in the comparison of step counts between rs1144566 genotypes no longer being “statistically significant” (new *P*-value = 0.314). Additionally, we have computed Cohen's *d* and confidence intervals for all two-sample comparisons, which are stated in the results section of the text (lines 264, 318, and 320).

We have updated the methods, text, figures, and figure legends to reflect these changes.

The “Statistics” section of the methods now reads (line 214):

“Comparisons of continuous variables between two groups were made using a two-tailed independent sample Welch's t-test for normally-distributed data or Mann-Whitney U test for non-normally distributed data in *scipy* v1.11.2 (Virtanen

et al., 2020). Determination of normality was made by visual inspection of data distributions and Q-Q plots generated in *scipy*. Effect sizes and confidence intervals between two groups were estimated by Cohen’s *d* and calculated using the Python module *pingouin* v0.5.5.”

Comment 10: The sine curve fitting methodology excludes approximately 35% of participants using an R^2 threshold of 0.5, which is not clearly justified. Sensitivity analyses using alternative thresholds and comparison of demographic characteristics between included and excluded participants would strengthen confidence that selection bias has not been introduced.

Response 10: We thank the reviewer for this crucial point and regret the absence of justification in our initial submission. In selecting our R^2 threshold, our objective was to balance goodness of fit and participant retention. We now show the relationship between these two factors in a new panel, **Figure 1B**. Additionally, we demonstrate that when goodness of fit and participant retention are considered equally, $R^2 > 0.5$ provides the optimal threshold based on Youden’s index. We have also updated the flow diagram in **Figure 1A** to reflect the fact that ~16% of participants had an $R^2 \leq 0.0$. Included below in **Reviewer Table 1**, we show odds ratios and adjusted *P*-values at R^2 thresholds of 0.4, 0.5, and 0.6 for type 2 diabetes mellitus-related diagnoses which were statistically significant in the manuscript PheWAS (using $R^2 > 0.5$).

We have updated the “Phase calculation” section of the methods to state (line 127):

“Optimal R^2 threshold was determined by the maximum Youden index which we computed as the participant retention rate plus the R^2 threshold minus 1; the participant retention rate was defined as the number of participants in the Fitbit cohort at a given R^2 threshold divided by the number of participants in the Fitbit cohort at an R^2 threshold of 0.0.”

We have updated the “Participant selection and background characteristics” section of the results to state (line 230):

“...the threshold of $R^2 > 0.5$ was selected using Youden’s index to balance dual interests in participant retention and goodness of fit (**Figure 1B**).”

Reviewer Table 1. Odds ratios of diabetes-related conditions at varying R^2 thresholds. Adjusted *P*-values were computed by the Bonferroni method. OR: odds ratio; CI: confidence interval.

Diagnosis Name	$R^2 > 0.4$			$R^2 > 0.5$			$R^2 > 0.6$		
	N affected (N=17,512)	OR [95% CI]	Adj. P-val	N affected (N=15,960)	OR [95% CI]	Adj. P-val	N affected (N=13,508)	OR [95% CI]	Adj. P-val
Type 2 diabetes mellitus without complication	2,318	1.09 [1.05,1.12]	0.0004	2,102	1.10 [1.06,1.14]	0.0002	1,799	1.09 [1.05,1.13]	0.0250
Diabetes mellitus without complication	2,372	1.09 [1.05,1.12]	0.0008	2,152	1.10 [1.06,1.13]	0.0004	1,842	1.09 [1.05,1.13]	0.0500
Type 2 diabetes mellitus	2,390	1.08 [1.05,1.12]	0.0012	2,161	1.09 [1.06,1.13]	0.0005	1,851	1.09 [1.05,1.13]	0.0390

Comment 11: Regarding the Mendelian randomisation analysis, I have limited experience with this form of analysis, but my understanding is that there should be mention of instrument strength metrics such as the F-statistic (ideally >10) or conditional F if adjusted for covariates. The authors acknowledge limitations of the single-SNV approach but may wish to expand on instrument validity.

Response 11: We apologize for any confusion regarding instrument strength metrics for the Mendelian randomization analyses. F-statistics and R^2 values for stage 1 of the 2-stage residual inclusion approach are included in the supplemental tables (**Tables S11, S12, S20, and S21**). We have now also included these values in the associated figure panels (**Figure 6B-C**).

Comment 12: Please ensure adherence to the Journal's policy on 'n' values: All 'n' values must be clearly stated main text, figures and their legends or tables with 'n' clearly defined (e.g. x cells from y slices in z animals) in each location. Authors should be mindful of how 'n' is defined to avoid pseudoreplication.

Response 12: We thank the reviewer for this point. We have now updated the text, figures, and legends to include all 'n' values. In addition, to ensure clarity and consistency, we have limited all analyses in **Figures 3-6** to the EHR+WGS cohort (n=15,960 participants). This change applies to **Figure 3A-C** and **Figure 5A-B** only, such that these panels now include the same cohorts as the accompanying panels in the same figures.

Dear Dr Kelly,

Re: JP-RP-2026-290337R1 "**Heart rate phase is an indicator of chronotype-relevant circadian shifts associated with human disease: An All of Us research program analysis**" by Zachary M Chan, Priya P Patel, Gabriella M Levitsky, Colleen A McClung, Seyed Mehdi Nouraie, and Neil J Kelly

We are pleased to tell you that your paper has been accepted for publication in The Journal of Physiology.

Yours sincerely,

Karyn Hamilton
Senior Editor
The Journal of Physiology

IMPORTANT POINTS TO NOTE FOLLOWING ACCEPTANCE OF YOUR PAPER:

- **IMPORTANT NOTICE ABOUT OPEN ACCESS:** To assist authors whose funding agencies mandate immediate public access to published research findings, The Journal of Physiology allows authors to pay an Open Access (OA) fee to have their papers made freely available immediately on publication.

The Corresponding Author will receive an email from Wiley with details on how to register or log in to Wiley Authors where you will be able to place an order.

- You can check if your funder or institution has a Wiley Open Access Account here:
<https://authors.wiley.com/author-resources/Journal-Authors/open-access/author-compliance-tool.html>

- You can help your research get the attention it deserves! Check out Wiley's free Promotion Guide for best-practice recommendations for promoting your work at: www.wileyauthors.com/eoo/guide. You can learn more about Wiley Editing Services which offers professional video, design, and writing services to create shareable video abstracts, infographics, conference posters, lay summaries, and research news stories for your research at: www.wileyauthors.com/eoo/promotion.

- If you would like to receive our 'Research Roundup', a monthly newsletter highlighting the cutting-edge research published in The Physiological Society's family of journals (The Journal of Physiology, Experimental Physiology, Physiological Reports, The Journal of Nutritional Physiology and The Journal of Precision Medicine: Health and Disease), please click this link, fill in your name and email address and select 'Research Roundup':
<https://www.physoc.org/journals-and-media/membernews>

EDITOR COMMENTS

Reviewing Editor:

No further comments.

Senior Editor:

Thank you for submitting your revised manuscript for continued consideration by The Journal of Physiology. The Referees were complimentary about the improvements resulting from the revisions. We are pleased to accept your manuscript for publication and appreciate your interest in The Journal of Physiology. Congratulations!

REFEREE COMMENTS

Referee #2:

No concerns at this point in time.

Referee #3:

I am satisfied that the authors have thoroughly and carefully addressed my comments. The statistical methodology has been appropriately revised, with the adoption of Welch's t-test, Mann-Whitney U tests, and the reporting of effect sizes and confidence intervals throughout. The justification of the R^2 threshold using Youden's index, together with the sensitivity analyses at alternative thresholds, adequately addresses the concern regarding potential selection bias. The consistent reporting of n values across figures and text is also appreciated. I have no further comments and am happy to recommend this manuscript for publication.